# Predicting fertility from sperm motility landscapes

Pol Fernández-López [1], Joan Garriga[1], Isabel Casas[2,3], Marc Yeste [2,3,4] & Frederic Bartumeus [1,4,5] ✉

Understanding the organisational principles of sperm motility has both evolutionary and applied impact. The emergence of computer aided systems in this field came with the promise of automated quantification and classification, potentially improving our understanding of the determinants of reproductive success. Yet, nowadays the relationship between sperm variability and fertility remains unclear. Here, we characterize pig sperm motility using t-SNE, an embedding method adequate to study behavioural variability. T-SNE reveals a hierarchical organization of sperm motility across ejaculates and individuals, enabling accurate fertility predictions by means of Bayesian logistic regression. Our results show that sperm motility features, like high-speed and straight-lined motion, correlate positively with fertility and are more relevant than other sources of variability. We propose the combined use of embedding methods with Bayesian inference frameworks in order to achieve a better understanding of the relationship between fertility and sperm motility in animals, including humans.

[1] Theoretical and Computational Ecology Group, Centre d'Estudis Avançats de Blanes (CEAB-CSIC), Cala Sant Francesc, 14, 17300 Blanes, Spain. [2] Biotechnology of Animal and Human Reproduction (TechnoSperm), Institute of Food and Agricultural Technology, University of Girona, 17003 Girona, Spain. [3] Unit of Cell Biology, Department of Biology, Faculty of Sciences, University of Girona, 17003 Girona, Spain. [4] Institució Catalana de Recerca i Estudis Avançats, ICREA, Passeig Lluís Companys, 23, 08010 Barcelona, Spain. [5] Centre de Recerca Ecològica i Aplicacions Forestals (CREAF), Cerdanyola del Vallès, 08193 Barcelona, Spain. ✉email: fbartu@ceab.csic.es

Gametes are the fundamental transferable units carrying the genetic variability and information that drive the evolution of populations, and ultimately, the survival of species. In farm animals, such as pigs or cattle, the information content in gametes is adjusted to millions of years of selective pressure conducted by breeders and, in more modern times, artificial insemination centres[1]. This pressure is targeted to augment the fertility of animals[2], and therein meat production, potentially ignoring other traits.

To accurately predict fertility outcomes, numerous sperm characteristics are examined[3–5], being sperm motility one of the most frequent[6–8]. In the attempt to make quantitative and objective measures of motility, Computer-Assisted Sperm Analysis (CASA) systems have played a prominent role[7,9–11]. CASA algorithms quantify different parameters in single sperm cells, providing big datasets of sperm motile properties[12].

The ability to move, from single cells to complex organisms, impacts on encounter success in a wide range of ecological contexts: food search, mating, reproduction, and species dispersal amongst others[13]. The motile capability of sperm, one of the few cell types common in all mammals, is expected to be shaped at optimising oocyte encounter and fertilisation. Many studies have tried to unravel the underlying physiological mechanisms and the relevance of sperm motility in reproductive success. However, the relationship between sperm motile properties and fertilising ability remains unclear. While literature points out some association between motility parameters and fertility (e.g., ref. [14–17]), most reports acknowledge poor predictability and practical application. This is so by a number of reasons. First of all, fertility depends on a number of interrelated factors other than sperm motility, where full genotype to phenotype individual mapping might be needed. Second, sperm motility is in itself highly variable, and characterising such baseline variability is not an easy task. Moreover, sperm undergo capacitation in the oviduct, a process that transforms several phenotypic aspects of sperm, including changes in motility patterns[18]. Only a capacitated spermatozoon is able to fertilise an oocyte, but it is difficult to mimic the oviduct conditions in the laboratory. Another big challenge is that fertility is not uniquely defined. Some of the commonly used traits to characterise fertility are: (i) the offspring at each delivery (litter size)[19,20], (ii) the proportion of inseminated females that do not return to the oestrus (non-return rate)[21–23], (iii) the proportion of inseminated females that become pregnant (conception rate)[24] and (iv) the proportion of inseminated females that reach farrowing (farrowing rate, FR)[16]. Finally, the selective pressure that farms and artificial insemination centres exert on animals clearly limits our ability to understand the role of natural sperm motility variation in overall fertility for such productive species[25].

In the present study, we shed light on the relationship between fertility and sperm motility, using the pig as a model organism. We focused on two main aspects: the need for a proper characterisation of sperm motility, and the need for a proper estimation of fertility, using the farrowing rate as metrics. To that extent, we propose: (i) a multivariate and discrete characterisation of the sperm movement to determine elementary behavioural modes or stereotypes[26–28], and (ii) a Bayesian multi-level regression framework to model determinants of fertility variation, and compare motility behavioural modes with other potential sources of sperm motility variation, which in turn, requires a proper assessment of the uncertainty associated to both regression coefficients and fertility estimates.

## Results

**High variability in boar sperm motility.** Sperm were characterised based on four motile-related properties: Curvilinear

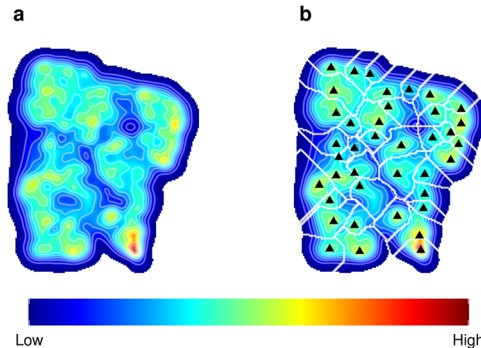

**Fig. 1 Sperm motility landscape resulting from a (Barnes-Hut approximation) t-SNE dimensional reduction. a** Kernel density showing the high- and low-density regions within the landscape, computed in a 200 × 200 cell grid with a neighbouring parameter representing a 1% of the dataset size (perplexity = 639). **b** Clustering using a watershed algorithm allowing the discretization of data into clusters (delimited by the white lines), and depicting the highest density peak within each cluster (black triangles). The legend describes the colour gradient in the density kernel. This analysis involves functions "bdm.pakde" and "bdm.wtt" in the bigMap R package. See also "Sperm motility landscape" in Methods and Garriga & Bartumeus 2018[75]).

velocity (VCL), straight-line velocity (VSL), amplitude of lateral head displacement (ALH) and beat-cross frequency (BCF). These are parameters that describe different aspects of sperm movement, and in principle, do not correlate or depend on one another. Using these four variables, we used a t-SNE algorithm to depict a bi-dimensional landscape where similar motile properties were packed together, conforming high-density peaks. In Figure S1 (Supplementary Note 1), we display the features that were used to characterise sperm motility.

Briefly, t-SNE computes pairwise distances in multivariate space, and represents the data in a 2D embedded map responding to one fundamental principle: similar behaviours should be close together. Accordingly, the closer the values of the four variables describing sperm motion are, the closer these motility behaviours are to one another, and thus, the closer they are found in the 2D embedding. In consequence, this two-dimensional space conforms to a complex landscape of peaks and valleys, where high densities of points evidence similar behaviours that are observed frequently (i.e., behavioural modes or stereotypes), whereas low-density areas depict uncommon behaviours. In order to unravel this structure, we computed a kernel of density over the position coordinates of this space (Fig. 1a) and discretized the landscape into behavioural clusters, using a watershed algorithm (Fig. 1b). As a result of this whole procedure, the motion patterns of more than 63,000 sperm cells' were grouped into 39 clusters.

The highly heterogeneous distribution of sperm motility patterns (Fig. 1) shows the existence of stereotypical behaviours in between a large space of variability. Stereotypes are sperm motility patterns that are recurrent and more abundant than others. Therefore, these are more likely to have important biological meaning.

We can map (Fig. S2 and Supplementary Note 2) the values of the four motility features (VCL, VSL, ALH and BCF) in the embedded motility landscape. In Fig. S2c we also show other variables that are derivatives of the four used to build up the landscape. This visualisation allows us to identify regions of well-differentiated behaviours (e.g., where are the faster sperm located) and some global patterns for each input feature (e.g., VSL mainly follows a vertical trend, with low values at the bottom of the landscape and high values at the top).

Interestingly, VCL and ALH shared quite similar distributions, while VSL and BCF were distributed differently, pointing out that VCL and ALH are somehow related.

One of the strengths of t-SNE methods is obtaining a comprehensive multivariate visualisation, that offers a complete perspective across scales of all the information contained in the landscape. We provide here a clear example by contrasting a t-SNE approximation to the supervised bi-variate sperm classification of CASA based on average path velocity (VAP) and straightness (STR = $100 \times$ VAP/VSL). Figure S2b shows that CASA sperm motility classes match almost perfectly with VAP patterns, and therefore, explain a limited diversity of the motility variability found in the population.

Sources of variability in sperm motility exist at different levels: among cells (spermatozoon), among ejaculates (from the same individual), and among individuals[29]. Some examples of this inter- and intra-individual variability are depicted in Fig. S3 (Supplementary Note 2). In nature, inter-individual diversity at genotypic and phenotypic levels usually defines individual fitness, and acts as the pool for further adaptation and survival improvement. However, in the case of sperm, intra-individual variability may also be important for fitness. Interestingly, we observed that the sperm collected from a particular boar on different days (different ejaculates) showed the same amount of variation in terms of motility patterns as the one observed between individuals; in addition, some males had much more restricted sperm motility landscapes than others. This could mean that no particular sperm behaviour can be attributed to a given individual, but rather boars share a spectrum of sperm motility patterns that may have separate functions in the ejaculate, including their suitability for fertilisation.

**Sperm motility influences fertility**. A simple approach to assess fertility is to consider the proportion of successes versus the number of attempts. This quotient is known as the farrowing rate (FR) and corresponds to the number of successful delivery (farrowing) with respect to the times that mating (or insemination) has occurred. Notwithstanding, this estimate is sensitive to sampling efforts and can be misleading and overrate or underrate "true" fertility (e.g., 1 single success in 1 attempt would be interpreted as a 100% chances of success). Consequently, we used Bayesian logistic regressions to estimate individual fertility as the probability of success per reproductive event. Such framework provided us with median fertility estimates, i.e., unbiased statistical estimates of the FR, and their associated credible intervals, controlling for sampling effort by adding each boar as a random effect.

We conducted a series of models, from simple to complex, assessing different sources of variability with potential impact on fertility (see Methods, Section "Modelling Fertility"). We first explored how intra- and inter-variability at the boar level, and sow parity (number of pregnancies per sow) affected the predictability of boar fertility (Table 1, models M0-M2). These three models (M0-M2) represented simple estimations of boar fertility that do not require sperm motility data, but are likely an accurate estimate of a FR. Model M3 (Table 1) incorporates the information of sperm motility per each of the boars, as the relative proportions of sperm distributed in 11 motility clusters. Indeed, we evaluated the predictive capabilities of model M3 with different landscape partitions (see Method Section "Modelling fertility" and Fig. 6). We tested an informative subset of landscape configurations (namely from 15 to two clusters) being the landscape with 11 clusters the one with the largest predictive capability, i.e., ELPD, expected log pointwise predictive density.

Our results (Table 1) show that adding information on sow parity and sperm motility (model M3) appears to be increasing the predictive accuracy (i.e., larger *ELPD*). On the contrary, the

**Table 1 Model comparison.**

| Model | Boar | Sow parity | Ejaculate | Motility | ELPD difference | SE |
|---|---|---|---|---|---|---|
| M0 | x | - | - | - | −2.8 | 3.0 |
| M1 | x | x | - | - | −1.9 | 1.9 |
| M2 | x | x | x | - | −2.6 | 1.9 |
| M3 | x | x | - | x | 0 | 0 |

A series of models were constructed as described in methods, "Fertility modelling" section. Null models (M0, M1, and M2, without motility information) were compared to models based on different landscape configurations, M3 being selected with an 11-cluster configuration, using an approximation of leave-one-out cross-validation and relative predictive performances (ELPDs) as scoring metrics (ELPD = 0 corresponds to the highest predictive ability). The model parameters are: Boar = inter-boar variability (as random effect), Sow parity = number of cycles of pregnancy of the sow, Ejaculate = intra-boar variability (as random effect), Motility = proportions of sperm in motility clusters, and the proportion of static spermatozoa of each boar. The last column (SE) shows the standard error of the ELPD difference, relative to the model with the highest ELPD (M3). See details about ELPD in loo R package documentation and Vehtari et al, 2017[79].

ejaculate variation does not seem to be adding significant information to the models ($ELPD_{M2} < ELPD_{M1}$). However, the differences between all the models are small, and not too relevant when considering the corresponding errors. We selected model M3 for further analysis and to assess which predictors in the model had a higher influence in FR.

In order to evaluate the impact of each parameter in the model, both coefficient values and uncertainties need to be taken into account. Coefficients close to zero, or with large overlap across zero are much less meaningful.

Contrarily, coefficients whose 50% credible intervals are on entirely positive or negative regimes are expected to be more relevant and robust, as a greater portion of their distribution surpasses the zero value. Although a positive contribution of the sow parity coefficient was observed (Fig. 2a), the major effects in the model were due to the 11 sperm motility clusters (Fig. 2b). Moreover, one should note that male variability was not much relevant, as the intercepts (and their distributions) showed little variation around zero (Fig. 2a). This is consistent with the results shown in Table 1, in which both sow parity and sperm motility behaviours appear to improve model performance.

Our results showed that sperm motility clusters could be either positively or negatively associated with fertility (Fig. 2b). We assessed which general sperm motility variables were characteristic of each cluster in order to infer what sperm motility behaviours may correlate positively or negatively with fertility. Broadly, larger values for VCL, VSL, ALH, VAP, LIN, STR and WOB were associated with reproductive success (Fig. 3a). Contrarily, larger values of BCF had a negative correlation to fertility. Even though all differences reported here exhibited a *p* value $< 10^{-3}$, not all of them were relevant to the same extent. Namely, ALH, VAP, STR, LIN, and VSL, had differences that ranged from 1.2 to 1.5 fold, whilst BCF, WOB and VCL displayed <1.2-fold differences.

We also colour-depicted the landscape, according to regression coefficient signs (Fig. 3b). This visualisation divides the landscape in two regions: the upper left region corresponds to clusters that have positive regression coefficients (positive impact on fertility), and the bottom right clusters have negative regression coefficients, thus, a negative effect on fertility. Clusters four and eight break this pattern, as their sperm motility features show a positive and a negative impact on fertility, respectively (Figs. 2b and 3b). A more detailed (per cluster) analysis is shown in Supplementary Note 3 and Fig. S4.

Finally, we assessed the overall predictive performance of the model. We compared the FR of each animal with the fertility estimates obtained from the model (Fig. 4). Most of the males

exhibited similar predicted fertility and FR, with slight differences both in values and ranking (average Kendall's tau ≈ 0.826, $p < 10^{-5}$ under 1000 permutations, Fig. 4a). Despite the high similarity between FRs and the predicted estimates, the later allow

quantifying the uncertainty of our predictions (Fig. 4b), providing fair information about the reliability of the estimated fertility, crucial in statistical inference[30,31].

Our results suggest the presence of three main groups: (i) boars with high fertility rate estimates (above 0.9) (positions 1–7); (ii) boars with intermediate fertility rates (between 0.8 and 0.9) and higher uncertainty (positions 8–13); (iii) boars with low fertility rates (below 0.8) and high uncertainty in the estimates (positions 14–17).

Both the relatively high uncertainties in some of the individual fertility predictions, and the similar fertility estimates of some of the boars, can affect the overall ranking (Fig. 4b). This is due to differences in sampling from the posterior distribution when predicting. For this reason, two measures are presented along with the estimated fertility to assess their robustness (Fig. 4a): a pseudo-probability of a particular boar to be ranked in a particular position, and the entropy of these probabilities (diversity in ranking of each boar). These two measures are key for the interpretation of the results. For instance, Boars 11 and 5 (Fig. 4a) showed a relatively low probability of being ranked in the second and third positions, respectively. However, the entropy in these particular boars is low, meaning that the range of positions in which they could be ranked is small, likely comprised between the second and the fifth position. In general, boars in the middle of the ranking (i.e., from positions 7–14) were found to be more sensitive to this variation (lower probabilities and higher entropies) than boars in the top or the bottom of the ranking.

We also tested whether the whole protocol of analysis was robust across different dimension reduction methods (see Supplementary Note 5). As depicted in Figs. S6 and S7, the results did not change qualitatively. While the ranking obtained through each method had some variations (most likely due to the large prediction uncertainties), the same boars were consistently found either at the top or at the bottom of the ranking regardless of the method, and were ranked in a similar position (Kendall correlations: $cor(BH|FIT) = 0.779$, $p$

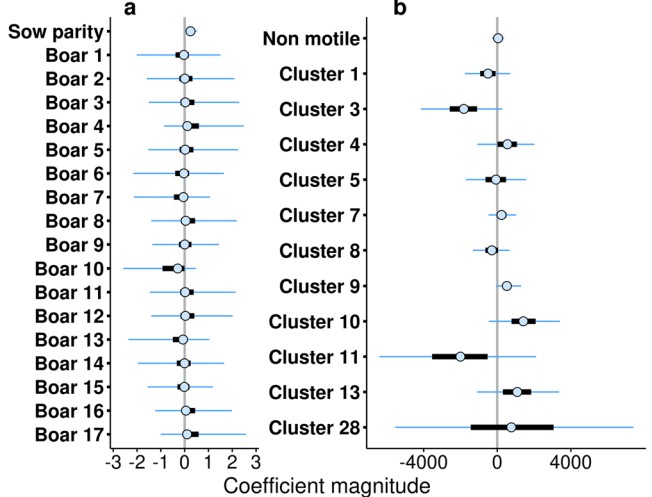

**Fig. 2 Estimates of the coefficients and their uncertainty in the fertility model.** The medians of the coefficients are depicted with a clear point, thick black bars represent the 50% credible intervals (C.I.), and the thin lines represent 95% C.I. **a** Coefficients of animal-related factors (sow parity, individual boars). Sow parity corresponds to the number of times that the sow completed a cycle of insemination. The boars (from 1 to 17) are intercepts that vary amongst individuals (usually known as random effect). **b** Coefficients corresponding to sperm behaviour related factors (the proportion of non-motile sperm, and the relative proportions of sperm in each of the motility clusters represented in the landscape). Note the different values in the x-axis (coefficient magnitude) for **a**, **b**.

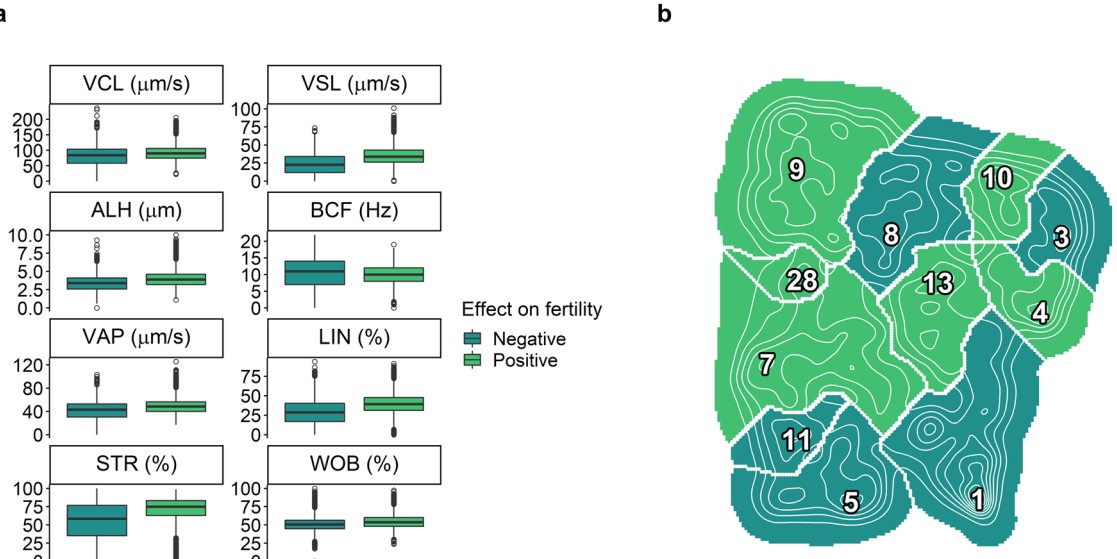

**Fig. 3 Impact of sperm motility features on fertility. a** Boxplots of the motility features (VCL, curvilinear velocity; VSL, straight-line velocity; ALH lateral amplitude of head displacement; BCF, beat-cross frequency; VAP, average path velocity; LIN, linearity; STR, straightness; and WOB, wobble) grouped by their effect (either positive or negative) on fertility, according to model M3. Mean values (positive-negative) for each motility variable: ALH (3.94−3.26 μm), VAP (48.72−40.29 μm/s), STR (71.61−54.26%), LIN (39.60−28.49%), VSL (34.94−23.64 μm/s), BCF (10.03−10.14 Hz), WOB (54.50−48.52%) and VCL (90.80−79.19 μm/s). Mean comparisons between the two groups (two-sided unpaired T-student tests; number of samples per group: positive = 36482, negative = 32688) yielded a $p$ value $< 10^{-15}$ in all motility variables, except for BCF ($p$ value $< 10^{-3}$). Both groups met the assumptions of T-student test in all variables tested. **b** 11-cluster motility landscape showing the clusters with positive (light green) and negative (dark green) correlation with fertility, according to model M3. The high-density peaks (labelled with the cluster number) are also depicted, as well as contour lines to help visualise the stereotypes (as in Fig. 1).

**a**

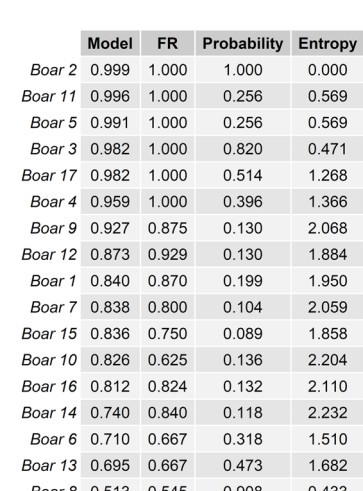

| Model | FR | Probability | Entropy |
|---|---|---|---|
| *Boar 2* | 0.999 | 1.000 | 1.000 | 0.000 |
| *Boar 11* | 0.996 | 1.000 | 0.256 | 0.569 |
| *Boar 5* | 0.991 | 1.000 | 0.256 | 0.569 |
| *Boar 3* | 0.982 | 1.000 | 0.820 | 0.471 |
| *Boar 17* | 0.982 | 1.000 | 0.514 | 1.268 |
| *Boar 4* | 0.959 | 1.000 | 0.396 | 1.366 |
| *Boar 9* | 0.927 | 0.875 | 0.130 | 2.068 |
| *Boar 12* | 0.873 | 0.929 | 0.130 | 1.884 |
| *Boar 1* | 0.840 | 0.870 | 0.199 | 1.950 |
| *Boar 7* | 0.838 | 0.800 | 0.104 | 2.059 |
| *Boar 15* | 0.836 | 0.750 | 0.089 | 1.858 |
| *Boar 10* | 0.826 | 0.625 | 0.136 | 2.204 |
| *Boar 16* | 0.812 | 0.824 | 0.132 | 2.110 |
| *Boar 14* | 0.740 | 0.840 | 0.118 | 2.232 |
| *Boar 6* | 0.710 | 0.667 | 0.318 | 1.510 |
| *Boar 13* | 0.695 | 0.667 | 0.473 | 1.682 |
| *Boar 8* | 0.513 | 0.545 | 0.908 | 0.433 |

**b**

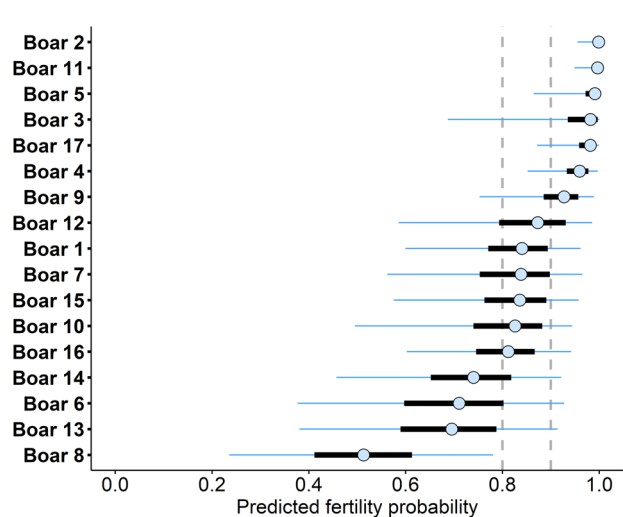

**Fig. 4 Predicted fertility per boar.** The values represent the median predicted fertility, expressed as probability of success (successful oocyte fertilisation). **a** Table of comparison between the estimated (posterior median) fertility, and the farrowing rate (FR), a commonly used measure of fertility. The boars are ranked in descending order according to the predictions of the model. The pseudo-probability (in a 1000 iteration bootstrap) is presented for each boar to be ranked in the current position. Additionally, entropy (calculated as $-\sum_{i=1}^{17} p_i \ln(p_i)$, where p is the probability of belonging to *i*th position) is used as a measure of position diversity for each of the boars. Namely, a higher entropy corresponds to boars that can be found in a broader range of positions, whereas a lower entropy corresponds to more robust boars. The Kendall's correlation between the predicted rank and the FR rank was calculated at each bootstrap iteration, yielding a mean correlation of 0.826 ± 0.055 (ranging from 0.662 to 1.000), with a mean *p* value of $1.7 \times 10^{-6} \pm 6.2 \times 10^{-6}$ (ranging from $5.5 \times 10^{-15}$ to $8.2 \times 10^{-5}$). **b** Predictions with their corresponding uncertainty around the median (open dot). 50% credible intervals (C.I.) are represented by thick, black lines, while 95% C.I. are represented with blue thin lines. Boars are sorted in descending order, according to their estimated fertility (median). The vertical dotted lines represent thresholds at 0.8 and 0.9, corresponding with the categorisation of boars (relatively low or high fertility, respectively).

value $= 1.3 \times 10^{-5}$; $cor(BH|UMAP) = 0.647$, *p* value $= 2.9 \times 10^{-4}$; $cor(FIT|UMAP) = 0.662$, *p* value $= 2.1 \times 10^{-4}$). This reinforces the presence of 3 groups of breeding males, as described above: (i) boars with high fertility rate estimates (above 0.9), (ii) boars with intermediate fertility rates (between 0.8 and 0.9) and higher uncertainty and (iii) boars with low fertility rates (below 0.8) and high uncertainty in the estimates.

**Capacitated sperm motility.** Sperm are ejaculated along with seminal plasma, which, amongst other functions, protect them from the harsh environment of the female reproductive tract[32]. As sperm progress in their journey towards the site of fertilisation, they encounter a gradient of molecules that modulate their behaviour. In addition, late capacitation events occur in the female oviduct (also known as Fallopian tube in humans) and consist of a series of physiological changes that modulate sperm function[33–35], including alterations in their motility patterns. Meanwhile in vivo imaging technologies are being developed and optimised[36], it is possible to mimic, in vitro, the biological context these sperm cells encounter along their journey through the oviduct. As capacitation is mandatory for a sperm cell to be able to fertilise the oocyte, we explored how this event transforms the behaviour of sperm in terms of motility.

We generated a dataset consisting of sperm capacitated in vitro. These new data accounted for 3851 motile spermatozoa, coming from nine different individuals. In order to evaluate whether or not new motility patterns were emerging in response to this process, we applied a similar t-SNE protocol to a combined dataset that included both capacitated and fresh sperm. A quick exploration of the data, showed that ~92% (3543 of 3851) of capacitated sperm represented new combinations of the four motility parameters assessed (VCL, VSL, ALH and BCF). That is,

most of the sperm in capacitating conditions exhibited new motile properties that were not present in the fresh sperm dataset.

In a deeper analysis, the behavioural landscape revealed that ~52% (1853 of 3543) of capacitated sperm were concentrated in a very delimited region, the island-like cluster at the top of the embedded landscape (Fig. 5). Indeed, this area represents extreme values in VSL (Fig. 5a), with an interquartile range from 89 µm/s to 139 µm/s, and maximum values up to 200 µm/s (not observed in fresh sperm). The other half of capacitated sperm were allocated mainly in three other regions; the top left corner, the middle left and the middle right (Fig. 5b, c).

Such areas of the landscape were characterised by low values of BCF (BCF < 10 Hz), and low values of VSL, especially in the more peripheral clusters (top left corner and middle right). Arguably, these could be non-capacitated sperm, that did not yet respond to the capacitating medium, or contrarily, responded too early and were thus found in a state of decay. Yet, in general, capacitated sperm accounted high-density peaks (behavioural stereotypes) in specific regions of the landscape where the density of fresh sperm was visibly lower or even null (Fig. 5c).

**Discussion**
A promise of the current behavioural data revolution and associated methods[37] is to generate new questions and shed light into old unsolved ones. Despite being explored experimentally for long time[38–40], it is still difficult to unravel the main drivers of reproductive success and, more specifically, knowing what is the role of sperm motility. At the root of this problem, there are very fundamental questions about how to quantify sperm motility or adequately address different sources of variability when making inferences between sperm motility and fertility. To gain knowledge on these issues, we have combined the use of large datasets,

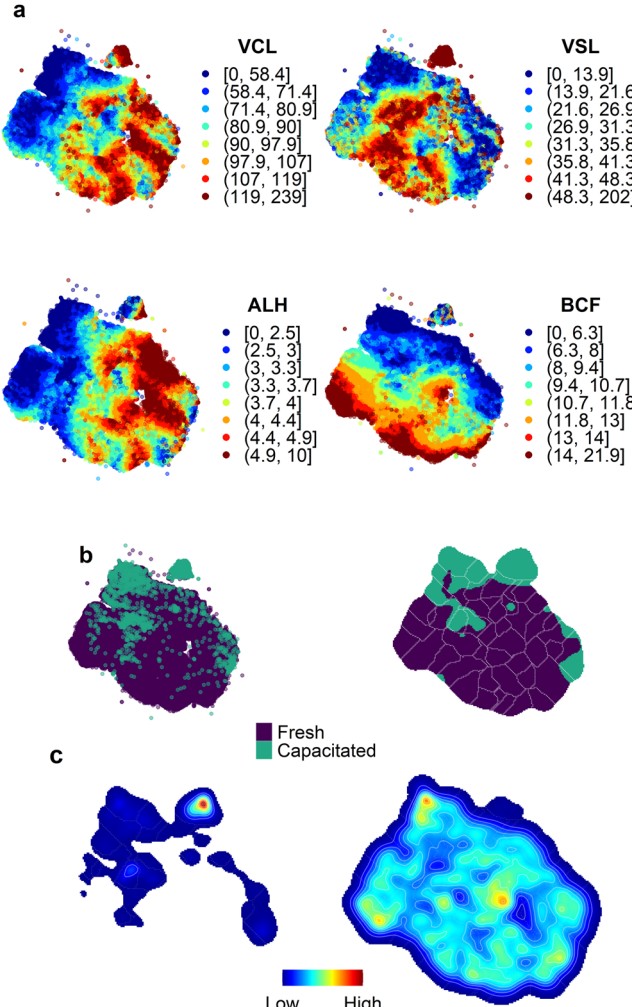

**Fig. 5 Sperm motility features of fresh and capacitated sperm. a** Quantile map of the motility features used in the t-SNE (VCL curvilinear velocity, VSL straight-line velocity, ALH lateral amplitude of the head displacement, and BCF beat-cross frequency). Colours represent an interval of values (eight percentiles, from 0–12.5 to 87.5–100) of each variable amended for a gradient visualisation. **b** Location of fresh sperm (purple) and capacitated sperm (green). Sperm labelled as capacitated were kept in a medium that elicits in vitro capacitation (containing albumin, bicarbonate and calcium, amongst others). The left panel is a pointwise representation of capacitated and fresh sperm. The right panel depicts a cell-wise hard clustering in a 200 × 200 grid. Cells were assigned "capacitated" or "fresh" depending on the density of points in each category (i.e., if a cell had a higher proportion of fresh sperm, it was labelled as fresh, and capacitated otherwise). **c** Heatmap of the landscape distribution of capacitated (left panel) and fresh sperm (right panel).

dimension reduction machine learning methods to characterise sperm motility behaviour, and Bayesian statistical inference.

Previous work on sperm motility used K-means and similar multivariate classification methods, leading from two to 11 clusters[4,17]. These studies put their focus on identifying sperm subpopulations and their possible biological functions. These methods are certainly a step forward in the characterisation of sperm behaviour. Yet, these methodologies have some limitations compared to t-SNE based and similar methods. Amongst them, (i) they offer limited options for visualisation, that is, exploration of comprehensive behavioural landscape across scales, (ii) some of them are supervised (i.e., a number of expected clusters must be specified as a parameter for the algorithm), and (iii) do not

explicitly consider behavioural modularity on the basis of similarity, i.e., motility stereotypes, as t-SNE (and other machine learning) methods do.

The presumption of motility stereotypes stems on a much broader view of behaviour as shaped by hierarchical and modular organisational principles[26,37]. The underlying concept is that behavioural variability comes in modules and is hierarchically organised. Such a view can be assessed comprehensively by using different perplexities and by merging clusters with some reasonable criteria (Garriga and Bartumeus, 2021, bigMap R package[41]). Choosing the right scale of observation is always difficult and depends on the scientific question or goal. Here, we have subordinated the choice of the scale of observation, and hence the number of clusters, to statistical inference (fertility predictions). As a potential drawback, statistically driven landscape partitions may limit their biological meaning or interpretability. Nonetheless, we found reasonable interpretations of the resulting landscape partition (the one that optimised the predictability of male fertility) enabling us to find specific motility features that contribute positively to fertility, and other than play the contrary role. These results are robust, as variations in our procedure (i.e., use of different machine learning methods to build the behavioural embeddings) ended up with qualitatively comparable landscapes, with 8–15 clusters presenting similar predictive capacities and sperm behavioural semantics.

Another key aspect is that the highly variable and multifactorial nature of reproductive success translates into large uncertainties in any fertility prediction. Notwithstanding, predictions are possible, and indeed, male fertility ranks obtained with Bayesian inference methods are not very different from crude FR quotients. The latter is often interpreted as a ground truth measure of fertility. However, the FR is just an estimate of fertility, also subject to biases as any other (e.g., due to sampling effort). The value of our method, in this case, is to generate a more accurate and less biased estimate of fertility[30,31], with a well-defined uncertainty associated with them. The approach suggested is valid in many other contexts in which behavioural landscapes may be used to infer other biological features.

In concordance with previous results[29], we have observed that sperm motility variation exists both at animal and ejaculate levels. Hence, different boars cover different regions of the overall sperm motility landscape, and ejaculates belonging to a particular boar exhibit the same range of variation as the one observed for different individuals. Large variability in sperm motility may be beneficial in contexts of constant change that demand adaptation both at individual and population levels. Because sperm encounter different types of obstacles in the female reproductive tract[42], greater sperm diversity could maximise the probabilities of overcoming them, increasing the chances of success. Our data suggest that, despite decades of artificial pig selection, some sperm behaviours promote better reproductive performance than others. Importantly, stereotyped behaviours (frequent and characteristic) occur in regions of the landscape with a positive and negative correlation to fertility. We identified that behavioural patterns producing linear and straight trajectories (mean LIN = 40%; mean STR = 72%) and high velocities (mean VCL = 91 µm/s; mean VSL = 35 µm/s; mean VAP = 49 µm/s) were related to more fertile boars. On the contrary, boar sperm cells showing lower and more variable speeds and less directional motion, correlated negatively with fertility.

Altogether, these results point out the existence of an optimal subpopulation of sperm within a much more heterogeneous pool of behaviours that are not related to oocyte fertilisation. Further studies to understand the role of the large variability observed in sperm motility, and in particular, the presence of suboptimal motility sperm behaviours, are much warranted. Of discussion is

whether suboptimal sperm could serve as the basis for further sperm motility selection[43,44], or else become relevant when combined with other external factors affecting reproductive success. For instance, there is evidence that the oviduct of the female responds to insemination at a transcriptomic level[45], pointing out the existence of adaptive mechanisms that could have a role in sperm selection.

Of note, we selected young males to maximise variability in FR, as old boars have already been selected based on their fertility. Hence it is possible that older boars (positively selected for high FRs) would exhibit less sperm variability, and a larger prevalence of stereotyped behaviours positively correlated with fertility. Similarly, it might well be the case that natural variability could be even much larger than the one observed in these young pigs. Comparing natural and artificially selected sperm motility patterns across ages, could provide useful information to understand which specific sperm behaviours have survived this selection in domestic animals, and potentially identify their role in reproduction.

Contrary to the diversity of behaviours that fresh sperm display, capacitation exerts a dramatic transforming effect, reducing this heterogeneity to a rather characteristic, unique behaviour. Although capacitation is known to shape sperm motion, there is a discrepancy in regards to what its actual effect is. While some authors have reported an increase in velocity, linearity and straightness[18,46], others claim a decrease in linearity paired with an increase in VCL[47].

We have observed two distinctive behaviours in sperm cells that were incubated in capacitating conditions. The most representative behaviour was characterised by extremely high values of VSL, in comparison to the distribution observed in fresh, uncapacitated sperm. The rest of the experimentally-conditioned sperm were localised in two areas of the landscape, both depicting low values of BCF, and mostly low VSL values. This latter behaviour is possibly due to differences in the sensitivity of each cell to experimental conditions[48], or because of the agglutination that occurs in the presence of $Ca^{2+}$[49] and prevents the free movement of the sperm. It is worth noting that both the "fast" and "slow" behaviours were located in the upper half of the landscape, in regions with the lowest range of BCF values. Although our characterisation of movement was made based on the head of sperm, this low BCF may correspond to the so-called vigorous flagellar beats: more strength in the flagellar movement seems necessary to achieve higher velocities with low frequency in the head movement (BCF).

According to our results, there is a notable distinction between fresh and capacitated sperm behaviours. These two differentiated patterns could indeed respond to two different needs or functions[50]. On the one hand, sperm cells have to reach the site of fertilisation (i.e., the oviduct). There are evidence that sperm transport along the female tract is aided by contractions of the myometrium[51–53]. This could relegate fresh sperm behaviour to a different role, which would not only be focused on reaching the oviduct, but also on a more plastic and diverse motion. Therefore, fresh sperm diversity could respond to a more elusive reactive behaviour, better at evading the immune system of the female, penetrating the cervical mucus, etc.

On the other hand, sperm motility would have a much more specific role in oocyte fertilisation, and their behaviour would change accordingly. For that purpose, sperm are known to be transported to a reservoir in a region of the oviduct[54–57], where they bind the epithelium[58,59] and await the arrival of the oocyte. Only then, capacitation gradually enhances the penetrating behaviour, shifting to the higher speed needed for sperm to release from the reservoir[60,61], penetrate the zona pellucida[62–64] and ultimately fertilise the oocyte.

Previous works have reported the existence of different factors that correlate to fertility in mammal species. Amongst them, sperm motility and morphology have been repeatedly studied to find easily measurable parameters that could be useful for fertility inference (e.g., [14–17]). These studies have observed a certain predictive value in sperm motility and morphology, characterising it through single-point estimates (i.e., mean values). In addition to these features, sperm DNA integrity has also been described as an important factor that (negatively) affects fertility[65–67]. DNA damage (as well as other non-apparent physiological defects) could indeed affect sperm functionality and behaviour. DNA integrity evaluation, however, is not as routinely performed as sperm motility and morphology analyses because it is more difficult to implement on a daily basis.

Behavioural landscapes seem more useful to study fertility than previously used methods, as the 2D embedding of sperm behaviour encompasses all the variability of the data, which indeed is hierarchically organised in behavioural modules. This allows us to study the relative prevalence of the different modes in explaining male fertility and how determinant sperm motility on fertility is, compared to other sources of variability (boar, ejaculate, sow parity). The predicted posterior distributions of male reproduction success revealed their estimated ratio according to our model, which turned out to be quite similar to FR, both in value and in ranking. While most boars showed small ranking differences between the model and the FR, some noticeable differences also existed (e.g., Boar 10). This could be due to either a higher uncertainty in these particular males, or in some of the regression coefficients. Likely, more data could improve the model and increase the accuracy of predictions.

In general, our model corroborated that most boars have high fertility ratios, despite our dataset consisting of young individuals where a decade-time artificial selection based on reproductive success was not expected to be the main driver (as it happens in old boars). This selective pressure on farm animals clearly limits access to animals with low fertility, hindering the possibility to obtain unbiased datasets. In any case, a proper estimation of the errors in both the coefficients and predictions is crucial to assess the potential limitations of the model and correctly interpret its accuracy. With the addition of sperm motility features, the uncertainties in the predictions diminished, resulting in a gain in the accuracy of the model.

This work represents a further step in understanding the complex interaction between sperm motility and fertility in a mammalian model. However, motility is only a part of a much wider network of factors. It is likely that some additional information at the cell level could shed some light on the mechanisms underlying the relationship between sperm behaviour and fertilisation success. Along these lines, coupling sperm motility with sperm morphology, or other information reflecting the cell's physiological state (such as DNA integrity) could provide some insight into the specific role of sperm motility behaviour. Furthermore, it could potentially reveal the importance of sperm motility stereotypes, and whether there is a relationship between the distinct motility behaviours and the internal state of the cell. It is also worth exploring the relationship between sperm motility and other factors indirectly related to fertility, such as sensitivity to capacitation. Remarkably, the analysis pipeline used in this paper is not restricted to sperm motility and could be applied to multivariate datasets obtained with flow cytometry (such as sperm chromatin structure assays) or other methods.

All in all, we developed a framework (i.e., characterisation of sperm behaviour, identification of stereotypes, and Bayesian inference modelling) that (i) is able to predict and reproduce other fertility estimates (i.e., FR), (ii) visualise the uncertainties associated to fertility (i.e., discriminate robust and weak

**Table 2 Components of the capacitating medium.**

| Component | Quantity (grams) |
|---|---|
| Hepes (buffer) | 0.1431 g |
| NaCl | 0.1965 g |
| KCl | 0.0069 g |
| Glucose | 0.0270 g |
| $Na_2HPO_4$ | 0.0033 g |
| $MgSO_4$ | 0.0030 g |
| $CaCl_2$ | 0.0198 g |
| Sodium lactate | 0.0729 g |
| Sodium pyruvate | 0.0033 g |
| Bovine serum albumin | 0.1452 g |
| Sodium bicarbonate (15 mM) | 0.0378 g |

The represented amounts are calculated to prepare 30 mL of medium. All reagents were purchased from Sigma Aldrich (Saint Louis, Missouri, USA).

estimation of fertility) and (iii) be further improved as new information becomes available (i.e., potential for decreasing uncertainty around the predictions). This framework could be applicable to several species, including humans, provided a dataset with multiple inseminations per male (i.e., as in reproduction clinics, where artificial insemination or in vitro fecundation are regularly practised).

## Methods

**Data collection and processing**. A first fertility dataset was provided by Batallé S.A. (Riudarenes, Girona). Only data comprised in the Spring period (from March to June) and corresponding to young boars (<13 months) were considered. This selection was meant to avoid possible seasonal and age effects. According to these criteria, we analysed a total of 36 ejaculates belonging to 17 different boars of Piétrain breed. Each ejaculate was used to inseminate from 1 to 25 sows, yielding from 4 to 32 total inseminations per boar. All animals were healthy and sexually mature, fed with a diet according to their nutritional requirements and provided with water ad libitum.

Sperm used to inseminate the sows previously passed different selection criteria. Namely, the ejaculates had at least a 70% sperm with normal morphology and motility. Progressive sperm motility was evaluated using a subjective scale, in which at least a score of 3 (in a scale from 0 to 5) was needed. The ejaculates passing these criteria were diluted in Duragen (Megapor SL, Zaragoza, Spain) and distributed in 45 mL insemination doses, at a concentration of 30 million spermatozoa/ mL. A total of three doses were used to inseminate (multiparous) sows. Sows that had not undergone parity (nulliparous) were inseminated with six doses instead.

Videos of fresh sperm (satisfying the aforementioned criteria) were recorded by Batallé S.A. as part of the usual procedure of semen quality analysis at the moment of semen collection. These videos were retrieved and further analysed by us to obtain individual motile properties for each spermatozoon, using ISAS software (Integrated Sperm Analysis System V1.0; Proiser SL, Valencia, Spain) with default settings.

Our dataset consisted of 98,020 spermatozoa, from which 28,850 were non-motile (not available motility parameters) and 69,170 were motile. For each spermatozoon, the available motility parameters were: (I) the area of the spermatozoon head (area, squared microns), (II) VCL (in μm/s), (III) straight-line velocity (VSL, in μm/s), (IV) averaged-path velocity (VAP, in μm/s), (V) linearity (LIN = 100 × VSL ÷ VCL, in %), (VI) straightness (STR = 100 × VSL ÷ VAP, in %), (VII) wobble (WOB = 100 × VAP ÷ VCL, in %), (VIII) amplitude of lateral head displacement (ALH, in μm) and (IX) beat-cross frequency (BCF, in Hz). Only motile sperm were considered to build the motility landscapes. Non-motile sperm were considered as proportion of static sperm per boar in the models.

A second dataset of sperm in capacitating conditions was obtained from other nine ejaculates belonging to nine different boars of the Piétrain breed, housed and fed under the same facilities and conditions than the boars in the first dataset. Ejaculates were transported refrigerated (15 °C) to TechnoSperm Laboratory, University of Girona (Girona, Spain), where sperm samples were incubated at 38.5 °C and 5% $CO_2$, in a medium containing bicarbonate, albumin, calcium and other molecules (Table 2)[68–70]. This medium is known to elicit sperm capacitation, and thus modulate their physiology. This dataset consisted of 3851 motile sperm cells.

**Sperm motility landscape**. In order to build a motility landscape from a set of sperm motile properties, we used an implementation of the t-distributed stochastic neighbouring embedding (t-SNE) algorithm that uses Barnes–Hut approximation to accelerate the computation of the landscape[71]. This functionality is present in

the bigMap R package (version 4.5.3, function "bdm.bhtsne"), available in github[72]. To explore the robustness of our results, we also used the Fast-Fourier Interpolation-based t-SNE (FIt-SNE)[73] and the Uniform Manifold Approximation and Projection (UMAP)[74] with the same dataset. We observed that, in qualitative terms, the results did not depend on the methods used to generate the motility landscape (Supplementary Note 5).

A matrix of 63,931 rows and four columns was used as input data. From the initial 98,020 sperm cells analysed, we excluded the non-motile ones (28,850 of 98,020, representing ~29.4 % of the dataset). After that, we removed the duplicated rows, which represented approximately a 7.5% of the motile sperm cells (5239 of 69,170 rows), as having redundant information in the dataset does not contribute to improving the landscape obtained by the t-SNE. Furthermore, duplicated points can strongly influence the forces of attraction and repulsion along the datapoints in the embedding. These forces are calculated based on the similarity between datapoints; because identical points have dissimilarity 0, they have a significant impact on the t-SNE landscape. This may result in a landscape organisation that revolves around the duplicated points, altering the structure of other regions in the embedding, and worsening the overall quality of the obtained landscape.

As for the columns, we used the four main motility parameters: VCL, VSL, ALH and BCF. Other parameters obtainable from ISAS software were excluded due to their high correlation with the previous variables or because they were quotients of the aforementioned features. In Fig. S1, we show a visual representation of these motility parameters to ease their interpretation.

The key steps to obtain the behavioural landscape are summarised as a sequential protocol using several functionalities of the bigMap R package[41,72] (https://github.com/jgarriga65/bigMap). For further information, refer to the documentation of bigMap R package and Garriga & Bartumeus (2018)[75].

Firstly, raw data were pre-processed ("bdm.data" function in bigMap with default values) through a principal components analysis (PCA) to homogenise ranges and weights. Afterwards, data were whitened to avoid unequal influence of the variables on the landscape. Then, we assessed a range of values on the critical parameter of the t-SNE algorithm: perplexity. Briefly, this parameter corresponds to the number of neighbours considered when computing pairwise similarities between datapoints. Low perplexities focus on accomplishing a landscape with better local structure, whereas high perplexities improve the global structure of the embedding[76]. We selected a perplexity of 639, corresponding to 1% of the dataset size, which was the best trade-off between local and global structure[41] (refer to Supplementary Note 4 and Fig. S5 for more details).

From the resulting 2D embedding, we computed an adaptive kernel density where the reference bandwidth parameter is also represented by perplexity, $KDE_{ppx}$ ("bdm.pakde" function in bigMap). This results in a point density heatmap showing high and low-density regions, corresponding to highly frequent (stereotypes) and non-frequent motile behaviours, respectively. This parameter ($KDE_{ppx}$) was set to 639 (1 % of the dataset size) in concordance with the perplexity used to build the landscape (t-SNE), in order to maintain consistent scales of analysis.

The last step, the clusterization of the landscape, was performed using the water track transform function in bigMap ("bdm.wtt"), a watershed algorithm that classifies behaviours based on density peaks.

The initial landscape discretization is highly-resolved, yielding from tens to hundreds of clusters. We can coarse-grain landscape information into a smaller number of broad clusters, less sensitive to initial conditions, through a hierarchical merging process that can be conducted on the initial clusters. Coarse-grained clusters incorporate more variability, but facilitate annotation and characterisation of the stereotyped, frequent behaviours. We used a hierarchical and recursive integration of clusters, according to a signal-to-noise ratio (S2NR) heuristic (function "bdm.s2nr.merge" in bigMap). Broadly, this computes each cluster variance vs. landscape variance ratio, to assess which cluster is the less informative in the embedding, and to which "parent" cluster it belongs in order to merge it. This heuristics can improve the signal-to-noise ratios in the overall landscape but also accounts for significant drops of information, when changes in cluster hierarchies occur and the landscape is strongly re-build to accommodate larger clusters. The hierarchical merging of clusters also allows identifying strongly preserved and frequent behaviours. The latter is represented as highly significant clusters that survive the coarse-graining process and attract the surrounding clusters. Each merged cluster encompasses the variability of a region of the landscape often ruled by one or a small subset of child clusters, the stereotypes, that represent to a good extent the behavioural features of the merged cluster.

**Modelling fertility**. To achieve a fair and robust estimate of reproduction success, measured as a fertility rate (usually called conception or farrowing rate), we used Bayesian multi-level logistic regression models. Broadly, logistic regressions are suitable to predict binary outcomes (i.e., success/failure), as is the case of FR estimates. On top of that, multi-level approaches are required to control for inter-individual variability (i.e., inter-boar variability). Multi-level logistic regressions can be implemented in Bayesian frameworks, which approximate the whole (predicted) posterior distribution of the outcomes, providing a better estimate than (95%) confidence intervals or single-point estimates. Furthermore, Bayesian frameworks allow introducing expected or known information (e.g., distribution shape, mean and dispersion) about the predictors in the model, potentially improving its

| Table 3 Parameters used for inference of boar fertility. | |
| --- | --- |
| **Parameter** | **Description** |
| Sow parity | The number of litters a sow has carried. This parameter introduces some information about the sow reproductive history in the model. |
| Boar | Boar variability at the individual level was included in the models as a level (random effect). This parameter represents non-controlled inter-boar variability. |
| Ejaculate | Ejaculate variability was included in the models as a level (random effect). This parameter represents non-controlled intra-boar variability. |
| Static sperm | Relative proportion of non-motile spermatozoa present in the ejaculates from boars. |
| Motile sperm | Relative proportions of spermatozoa (from each boar) on different motility clusters in the landscape. |

The distinct parameters explored to model boar fertility are presented, along with a brief description.

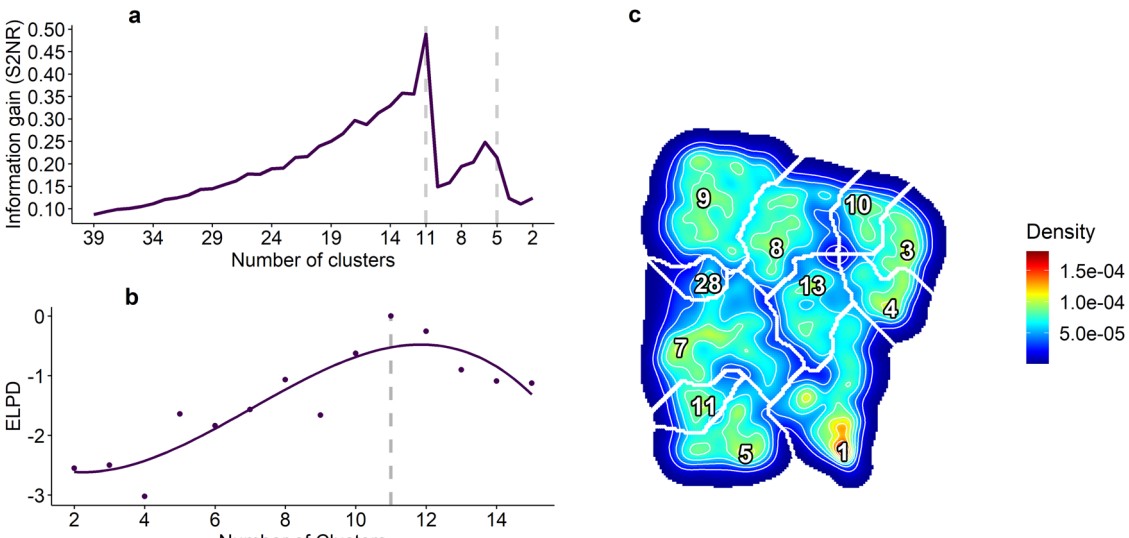

**Fig. 6 Model selection based on sperm motility landscape configurations. a** Signal-to noise ratio (S2NR) as an heuristic of information gain, along the process of merging. The information generally increases along the merging process. However, there were some drops, that corresponded with significant changes in the hierarchical organisation of the landscape. **b** Leave-one out method for cross-validation, used to assess model performance (loo R package, "loo" and "loo_compare" functions, see details in Vehtari et al., 2017[79]). The ELPD corresponds to the expected log pointwise predictive density, as a measure of predictive capability of the model. Models of different landscape configurations (merging scales) were compared, 11 clusters being the best partition for model performance (ELPD = 0). **c** Motility landscape with 11 merged clusters that encompass the variability of the whole landscape. This represented both a good compromise between information and interpretability, and a good model performance.

accuracy, as well as reducing the uncertainty associated with predictors and predictions.

We generated a set of Bayesian multi-level logistic regression models in a three-step analysis, using combinations of the parameters described in Table 3. To this purpose, we used rstanarm R package (version 2.21.1)[77,78]. Briefly, this package provides a great variety of modelling functions, model diagnosis and posterior distribution analysis. For details see "rstanarm" documentation.

As a first step, we built a null model (M0 = Boar), only containing male information, introduced as a level (random intercept) to obtain an estimate of fertility just based on boar variability. From this simple model, we sequentially added layers of complexity and assessed the influence on the predictability of each model. The next model included the boar information, as well as some information about the sow, namely the sow parity (M1 = Boar + Sow). The following model had, in addition to boar and sow information, the ejaculate information as a random intercept (M2 = Boar + Sow + Ejaculate). By comparing these three models (M0, M1 and M2), we observed that sow information improved predictions, whilst adding the ejaculate information had the opposite effect (worse predictability).

The next step consisted of adding the motility effect to the models. Because the best of the three models was the M1, we built a new model, including the boar and sow information, in addition to the motility features of sperm (M3 = Boar + Sow + Sperm motility). All models (M0 to M3) are summarised in Table 1 (results section), with their relative predictive ability (ELPD), all models exhibiting similar prediction capability (similar ELPD).

Sperm motility was introduced in model M3 as proportions of sperm in each motility cluster of the landscape, for each of the boars. As the landscape was highly complex (39 clusters), we first merged them one-by-one following a recursive, spatially hierarchical strategy based on signal-to-noise ratio (S2NR) heuristics (see

section above). As clusters were merged, the S2NR progressively augmented so that the overall clustering information increased over noise, but only up to certain levels where abrupt drops of information took place (when reaching 11 and five clusters). After these drops, subsequent merging reorganised the landscape and brought the S2NR back to larger values (Fig. 6a). This procedure allowed us to find levels of granularity in our landscape, which showed good enough statistical signal (i.e., inter/intra cluster variance) and a reasonable number of clusters for us to be able to interpret sperm motility modes and infer male fertility. Considering all this, we chose the subset of landscape partitions comprised between 15 and two clusters (Fig. 6a), which emerged as the most adequate behavioural scales in order to infer male fertility.

A logistic regression model was performed for each landscape configuration, (from two to 15 clusters) using "stan_glmer" function in rstanarm R package, with the following parameters: chains = 4; adapt_delta = 0.99; family = binomial(link = "logit"); iter = 4000; prior = normal(0, 5, autoscale = TRUE). For details, check "rstanarm" documentation and https://mc-stan.org/users/interfaces/rstan.

The models obtained at each of these scales were compared using a leave-one-out based ("loo" function in loo R package, version 2.3.1)[79,80] cross-validation method (Fig. 6b). Briefly, this method computes approximate leave-one-out cross-validation without the need of re-fitting the model with different training sets (see details in Vehtari et al.[79]). Our results showed that the largest ELPD (expected log pointwise predictive density) was obtained when including 11 sperm motility clusters (Fig. 6b), coinciding with the most informative configuration of the landscape (largest S2NR), right before a substantial change in the landscape hierarchy (drop in SN2R). In other words, this particular landscape partition (Fig. 6c) is highly interpretable and informative (low number of clusters and large S2NR) and is the best to infer boar fertility (large ELPD). Therefore, model version M3 (Table 1) contains the sperm motility behaviours at this particular scale.

**Statistics and reproducibility**. R software was used to perform all analysis, and the required packages and functions are cited in the corresponding sections of methods. The main pipeline of analysis consisted of (i) pre-processing data, (ii) building a behavioural landscape, (iii) discretizing the landscape in clusters and (iv) using the proportion of sperm in each cluster to model boar fertility. Reported results for the coefficients and predictions are represented as medians with their 50% and 95% associated credible intervals. Mean values for the motility features are described in the main (results) text. Figure legends include information about statistics used, and any possible descriptive metrics about means, errors and $p$ values. The data used in the figures are available in Mendeley Data (DOI: 10.17632/jd38jhxpg6.4)[81] and the corresponding code is available in github (https://github.com/Polfe94/sperm_move) and zenodo (https://doi.org/10.5281/zenodo.7015571)[82].

**Reporting summary**. Further information on research design is available in the Nature Research Reporting Summary linked to this article.

## Data availability

Fresh and capacitated sperm motility data and boar fertility data are available in a publicly accessible Mendeley Data repository (https://doi.org/10.17632/jd38jhxpg6.4)[81]. In the same repository, source data for the figures of this manuscript can also be found. Please check the related github repository and read the instructions about how to reproduce this work.

## Code availability

R and Python code used for building the behavioural landscape and Bayesian modelling is publicly available in a github repository (https://github.com/Polfe94/sperm_move) and zenodo (https://doi.org/10.5281/zenodo.7015571)[82]. Codes for reproducing the figures of the present manuscript are also available in the aforementioned repositories.

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

## Acknowledgements

We thank Selecció Batallé S.A. for providing the ejaculates and the fertility and motility data. In particular Josep Reixach for the exceptional job of gathering data that met the purposes of the study, and Sergi Linares for retrieving the videos of the sperm. We also would like to thank Marta Guitart for collecting the ejaculates used in the capacitation experiment. This work has been financially supported by grants no. CGL2016-78156-C2-1-R, AGL2017-88329R, RYC-2014-15581 and PEJ2018-002178-P (Ministry of Science, Spain), and 2017-SGR-1229 (Generalitat de Catalunya, Spain). Figure S1 was created using BioRender.com.

## Author contributions

F.B. and M.Y. designed the work strategy and main goal. I.C. contributed to database acquisition and initial data processing. J.G. contributed to t-SNE methods assessment and algorithmic development. P.F. executed all the data analysis guided by F.B. and wrote the first draft of the MS. All coauthors contributed to further versions of the text.

## Competing interests

The authors declare no competing interest.
