## [Peer Review File · Communications Biology]

Reviewers' comments:

Reviewer #1 (Remarks to the Author):

Predicting fertility from sperm motility landscapes

Reviewer Ann Van Soom

General comments

This paper described a specialized analytical method, t-distributed Stochastic Neighbour Embedding or T-SNE, which revealed a hierarchical organization of sperm motility across ejaculates and individuals (pigs), enabling fertility prediction by means of Bayesian logistic regression. The authors show that sperm motility features, like high-speed and straight-lined motion, correlate positively with fertility, and are more relevant than other sources of variability. This is an interesting observation, since this feature has been noticed already been noticed to be related with field fertility by technicians working with bull and dog semen, but was difficult to prove. Also other semen parameters have been related with field fertility, but are not discussed in the paper. The most important correlation besides motility to correlated with field fertility is DNA integrity (in pigs (<https://pubmed.ncbi.nlm.nih.gov/20458156/> and in bulls <https://pubmed.ncbi.nlm.nih.gov/11291917/>) as assessed by the sperm chromatin structure assay SCSA. This correlation was first published by Evenson in an old Science paper <https://pubmed.ncbi.nlm.nih.gov/7444440/> , leading to 135 papers by the same author on essentially the same subject in different species on Pubmed. So at least this relationship with field fertility should be mentioned as well.

I do not have sufficient understanding of statistics or artificial intelligence to review the methods used in this paper, plus corresponding results. I found the paper in general well-written and the experiments well -designed with correct interpretation of results. I do have however a few suggestions for improving the paper, to make it more accessible for scientists working with pig semen routinely. Although the paper is quite balanced between mathematics and biology, I believe that more information is needed on the biology side to make it more accessible for practitioners working with stud animals

Specific comments

Does the manuscript have technical or conceptual flaws that should prohibit its publication? I am not sufficiently acquainted with the methodology to assess this

Are the conclusions original? Yes but some more informaton is needed , conclusion is now too strong, please see below

Do you feel that the results presented are of immediate relevance for people in your own discipline or for a broader audience? They are indeed relevant for people working in AI but some more information is needed, especially in the material and methods, discussion and conclusions.

Material and methods

Data collection : please indicate the cut-off values used to approve the ejaculates of the boars. % Live, % normal morphology, % progressive motility, total sperm output. Also give more details on diluter composition, final concentration of diluted semen etc.

I could not find how many sows (min-max) were inseminated per boar/ejaculate, and what their farrowing rate/litter sizes were but maybe I missed it . I just found "first fertility set", without further explanation. In Figure 4 there were data on predicted fertility per boar but it was not clear to me how that correlated with actual fertility

Discussion

I understand that the focus of the paper is on sperm motility, but as mentioned before, I missed the importance of sperm morphology and DNA-integrity. Boars are selected for fertility, with strict criteria for sperm morphology, so all samples would contain a sufficient percentage of normal sperm cells but still DNA integrity may be affected without being detected by routine semen evaluation. Hence it is necessary to add a paragraph on this in the discussion.

Concluding- Better replace by conclusion or conclusive remarks

Here I suggest to also add that additional information at the cell level such as evaluation of DNA or chromatin integrity could be of benefit.

Final line : "We believe that a similar protocol could be applied to a broad range of species, including humans".

This is quite a bold statement, which I think is not completely right. Maybe you should add in the discussion that field fertility data and relation with semen quality are easy to obtain in pigs and cattle, because boars and bulls will be used for inseminating hundreds of females. To obtain field fertility data is already more difficult for people working with horses and dogs, where less males are being used on a lower number of females, and in humans it may even be not possible, because man father only a limited number of children with a single woman (in general). So a bit more background on this statement will be needed, or the statement has to be adapted. Or can the statistical method be translated to other species just like that? I do not think that is quite likely, but maybe I misunderstood.

References

I also noticed that the references were not numbered in the reference list, this should be corrected.

Reviewer #2 (Remarks to the Author):

The study aimed on two main aspects: the need for a proper characterization of sperm motility, and the need for a proper estimation of fertility, using the farrowing rate as metrics. In my opinion this is a classic example of a good paperwork. The whole manuscript is complete and intelligible. As a matter of fact, all chapters of mentioned draft have been written in detail. A sufficient number of animals as well as up-to-date analytical and statistical methods (multivariate and unsupervised methods) were used in the study. Presented documentation does not raise any objections and conclusion follows the results obtained by the authors.

Reviewer #3 (Remarks to the Author):

This paper studied how the heterogeneity of sperm motility is related to fertility. They used t-SNE embedding of sperm motility to build a Bayesian logistic regression model for the prediction of fertility. This is a relatively straightforward machine learning method. They claimed that they were able to accurately predict and rank male reproductive success, but there is no comparison with baseline or pre-existing models. Therefore, it is difficult to know how much advancement this method achieved. They also found that sperm motility features, like high-speed and straight-lined motion, correlate positively with fertility, which seems to be already expected. Therefore, it is not clear what new knowledge was acquired by this analysis. The followings are specific points that the author should consider to improve their manuscript.

- There is no systematic testing of their analysis pipeline. There are alternative methods for dimensional reduction such as PCA and UMAP, which should be tested as well. What is the rationale for the Bayesian logistic regression? What happens if a different machine learning method is used?
- There are many biological terminologies that are not clearly defined. For the readers who are not experts in this field but are interested in the method, the terminologies (e.g. boar, show parity, capacitation) should be clearly explained.
- The methodology was not clearly documented and it is difficult to follow it. The author should make a clearer explanation of the analysis pipeline.
- Training and testing procedures of the model are not clear. They mentioned leave-one-out cross-validation, but it is difficult to know how validation and testing were performed.
- Statistical testing is not well documented. How many samples? If t-test was used, did the data

follow Gaussian distribution?

- The prediction performance was measured by ELPD. But there is no mathematical definition of it. Also, there should be error bars of ELPD and their statistical testing in Table 1
- t-SNE generates different embedding when it is repeated. The author should assess the reproducibility of the model.

Reviewer #1

General comments

This paper described a specialized analytical method, t-distributed Stochastic Neighbour Embedding or T-SNE, which revealed a hierarchical organization of sperm motility across ejaculates and individuals (pigs), enabling fertility prediction by means of Bayesian logistic regression. The authors show that sperm motility features, like high-speed and straight-lined motion, correlate positively with fertility, and are more relevant than other sources of variability. This is an interesting observation, since this feature has been noticed already been noticed to be related with field fertility by technicians working with bull and dog semen, but was difficult to prove. Also other semen parameters have been related with field fertility, but are not discussed in the paper. The most important correlation besides motility to correlated with field fertility is DNA integrity (in pigs (<https://pubmed.ncbi.nlm.nih.gov/20458156/>) and in bulls (<https://pubmed.ncbi.nlm.nih.gov/11291917/>)) as assessed by the sperm chromatin structure assay SCSA. This correlation was first published by Evenson in an old Science paper (<https://pubmed.ncbi.nlm.nih.gov/7444440/>), leading to 135 papers by the same author on essentially the same subject in different species on Pubmed. So at least this relationship with field fertility should be mentioned as well.

R: According to the reviewer's advice, we have stated the importance of these findings (correlation between DNA integrity and individual fertility) in the manuscript (discussion section). Additionally, we have mentioned the potential insights DNA integrity and sperm motility and morphology could provide when analysed together, using the pipeline of analysis proposed in our paper.

I do not have sufficient understanding of statistics or artificial intelligence to review the methods used in this paper, plus corresponding results. I found the paper in general well-written and the experiments well -designed with correct interpretation of results. I do have however a few suggestions for improving the paper, to make it more accessible for scientists working with pig semen routinely. Although the paper is quite balanced between mathematics and biology, I believe that more information is needed on the biology side to make it more accessible for practitioners working with stud animals.

R: In concordance with the reviewer's criteria, we have improved some parts of the manuscript, extending the biological implications of our method (having into account the points mentioned in the specific comments below).

Specific comments

Material and methods

Data collection : please indicate the cut-off values used to approve the ejaculates of the boars. % Live, % normal morphology, % progressive motility, total sperm output. Also give more details on diluter composition, final concentration of diluted semen etc.

R: We acknowledge that this information was not clearly detailed in the manuscript. We have revised this section providing further information from the commercial farm (Batallé S.A.; lines 392-398)

I could not find how many sows (min-max) were inseminated per boar/ejaculate, and what their farrowing rate/litter sizes were but maybe I missed it. I just found “first fertility set”, without further explanation. In Figure 4 there were data on predicted fertility per boar but it was not clear to me how that correlated with actual fertility

R: As the reviewer suggests, we have added the number of inseminated sows per boar and ejaculate in the manuscript (lines 389-390). We have also confirmed that farrowing rates were already included in the previous version of Manuscript (Figure 4A; FR, second column of the table). Although litter size was not used as a measure of fertility in the present manuscript, they are publicly available (Reserved DOI: 10.17632/jd38jhxpg6.2). In the data set, there is detailed information about the age of boars, sow parity, number of piglets born (litter size), insemination date, and other information. The average litter size was 15.16, with an average of 1.26 stillborn piglets.

In regards to Figure 4, and the correlation between boar fertility and farrowing rate, we would like to clarify some aspects. The table displayed in Figure 4A does not show a correlation between predicted fertility and *actual* fertility. Rather, it shows the predicted fertility (according to the model) and the farrowing rate. As can be observed in the table (Fig. 4A), these two measures (Model, FR) yield similar estimated values of fertility. We discuss in the manuscript that farrowing rate should not be interpreted as *actual* fertility, but as another estimate of fertility (lines 112-114). To emphasize this idea, we have mentioned it again in the discussion (lines 266-268). We discuss that the main benefit of the model predictions is to have a fair estimation of the errors associated with the predicted fertility, whereas the farrowing rate is sensitive to sampling effort (i.e. it is strongly influenced by the number of inseminations). While there is a strong correlation between farrowing rate and the model's predictions ($R = 0.897$, $p < 10^{-5}$, 95% C.I. [0.733, 0.963]; calculated from the data on the table), we focus on fertility rankings, which are easier to compare than the actual estimates. Rankings are shown to be highly correlated (average Kendall's tau = 0.826), even though the predictions fluctuate based on the samples drawn from the predicted posterior distribution (i.e. there is a stochastic component in the predictions). The probability and entropy measures presented (Fig. 4A) are aimed to characterize these fluctuations, so the readers can evaluate the robustness of the ranking obtained with the model. All in all, farrowing rate is a simple and straightforward estimate of fertility. However, the model can: (i) reproduce its behaviour (predict / estimate boar fertility), (ii) visualize the errors or uncertainties associated to fertility (detect which boars have *robust* fertility estimates, and which ones exhibit *strong* fluctuations on their fertility estimate) and (iii) be further improved with new information, possibly decreasing the uncertainty around the predictions.

Discussion

I understand that the focus of the paper is on sperm motility, but as mentioned before, I missed the importance of sperm morphology and DNA-integrity. Boars are selected for fertility, with strict criteria for sperm morphology, so all samples would contain a sufficient percentage of normal sperm cells but still DNA integrity may be affected without being detected by routine semen evaluation. Hence it is necessary to add a paragraph on this in the discussion.

R: We understand the relevance of integral semen quality assessment (morphology, motility and others). Therefore, and in agreement with the reviewer, we have deepened more in the discussion about the need to consider all these factors in semen quality analysis (lines 335-344).

Concluding- Better replace by conclusion or conclusive remarks

R: We changed the title of the conclusions according to the reviewer's criterion.

Here I suggest to also add that additional information at the cell level such as evaluation of DNA or chromatin integrity could be of benefit.

R: We appreciate the suggestions of the reviewer. The corresponding information has been added in the discussion (lines 368-376).

Final line : “We believe that a similar protocol could be applied to a broad range of species, including humans”.

This is quite a bold statement, which I think is not completely right. Maybe you should add in the discussion that field fertility data and relation with semen quality are easy to obtain in pigs and cattle, because boars and bulls will be used for inseminating hundreds of females. To obtain field fertility data is already more difficult for people working with horses and dogs, where less males are being used on a lower number of females, and in humans it may even be not possible, because man father only a limited number of children with a single woman (in general). So a bit more background on this statement will be needed, or the statement has to be adapted. Or can the statistical method be translated to other species just like that? I do not think that is quite likely, but maybe I misunderstood.

R: We agree with the reviewer that fertility data are more difficult to be obtained in some species, like humans, and probably field data is unlikely to be available. However, in a reproduction clinic where IUI, IVF and ICSI are a regular praxis, these data should be somewhat easy to obtain. Semen donors would be a good target to study male fertility as we do in our manuscript with boars: the sperm of a single male are used to inseminate several females. In this regard, one could apply the same protocol (i.e. t-SNE, clusters, models), and estimate fertility as “Number of pregnant women (and successful delivery)” / “Number of total inseminations (from a given man)”, as we did with farrowing rate. We have, in concordance with the reviewer's perspective, adapted the statement so that this is reflected (lines 380-384).

References

I also noticed that the references were not numbered in the reference list, this should be corrected.

R: As the reviewer points out, references were not numbered. We have corrected this issue.

Reviewer #2

General comments

The study aimed on two main aspects: the need for a proper characterization of sperm motility, and the need for a proper estimation of fertility, using the farrowing rate as metrics. In my opinion this is a classic example of a good paperwork. The whole manuscript is complete and intelligible. As a matter of fact, all chapters of mentioned draft have been written in detail. A sufficient number of animals as well as up-to-date analytical and statistical methods (multivariate and unsupervised methods) were used in the study. Presented documentation does not raise any objections and conclusion follows the results obtained by the authors.

R: We are grateful for the reviewer's comments and perspective.

Reviewer #3

General comments

This paper studied how the heterogeneity of sperm motility is related to fertility. They used t-SNE embedding of sperm motility to build a Bayesian logistic regression model for the prediction of fertility. This is a relatively straightforward machine learning method. They claimed that they were able to accurately predict and rank male reproductive success, but there is no comparison with baseline or pre-existing models. Therefore, it is difficult to know how much advancement this method achieved. They also found that sperm motility features, like high-speed and straight-lined motion, correlate positively with fertility, which seems to be already expected. Therefore, it is not clear what new knowledge was acquired by this analysis. The followings are specific points that the author should consider to improve their manuscript.

R: As a general comment, cutting-edge statistical methods linking sperm behaviour and fertility metrics do not, to the best of the author's knowledge, exist. We have developed a new framework that can generate an optimized behavioural landscape that may explain fertility better. We do confirm some previously expected results (e.g. some motility metrics can favour fertility) but we do not consider this to be our main outcome (though is informative). As we answered to Reviewer #1, our modelling framework can: (i) reproduce boar fertility behaviour (predict / estimate boar fertility), (ii) visualize the errors or uncertainties associated to fertility predictability (detect which boars have *robust* fertility estimates, and which ones exhibit *strong* fluctuations on their fertility estimate) and (iii) be further improved with new information, possibly decreasing the uncertainty around the predictions.

Specific comments

- **There is no systematic testing of their analysis pipeline. There are alternative methods for dimensional reduction such as PCA and UMAP, which should be tested as well. What is the rationale for the Bayesian logistic regression? What happens if a different machine learning method is used?**

R: We would like to say that we do not share the reviewer's view, as there was indeed a systematic analysis of the pipeline in our initial submission. In the supplementary material section "robustness across t-SNE implementations" (initially submitted version) we presented the results obtained through two different t-SNE based methods.

Additionally, and following the reviewer's advice, we further tested the analysis pipeline with UMAP, which is a conceptually similar method to t-SNE. Results are presented in the same supplementary material section (renamed as "Robustness across dimension reduction algorithms"), Figs. S6 and S7.

Regarding the use of PCA, we believe that this method is not aligned with one of the main goals of the paper, which is to characterize sperm motility heterogeneity in terms of similarities and stereotyped behaviours. In Berman et al. 2014 (Discussion section) the reviewer can read the conceptual framework explaining why t-SNE methods can be insightful in behavioural analysis. PCA attempts a reduction of dimensions through variance maximization, and therefore, does not fit the purposes of our study. Having said that, we describe in methods section (lines 452-454) that PCA was performed as a data pre-processing step (variable scaling).

In order to make predictions, we used a Bayesian multi-level logistic regression. The use of logistic regression is suitable to predict binary outcomes (i.e. 1 / 0, success / failure, pregnancy / non pregnancy, ...). Multi-level approaches are required to control for inter-boar variability. Bayesian frameworks allow to obtain the whole (predicted) posterior distribution of the outcomes, providing a better estimate than a (95%) confidence interval or single point estimates such as the median or the average. Furthermore, Bayesian frameworks allow to modify the expected or known features of the distribution of the predictors in the model; these models can thus be further improved with the proper information (e.g. distribution shape, mean and dispersion). Altogether, Bayesian Multi-Level Logistic Regression offers the potential for describing the distribution of the predicted outcome, the potential for model improvement (as new knowledge is available for the predictors introduced in the models) and good characterization and visualization of the predicted outcomes and their associated errors. We acknowledge that the rationale of the modelling via Bayesian multi-level logistic regressions was not mentioned in the manuscript. We have added this information in a methods subsection ("Modelling fertility", lines 489-497)

- There are many biological terminologies that are not clearly defined. For the readers who are not experts in this field but are interested in the method, the terminologies (e.g. boar, show parity, capacitation) should be clearly explained.

R: We agree with the reviewer that some biological terms are specific and could be confusing for non-expert readers. We carefully reviewed the definitions of these concepts and corrected or extended them where necessary.

- The methodology was not clearly documented and it is difficult to follow it. The author should make a clearer explanation of the analysis pipeline.

R: We are willing to accept that the methodology used in the manuscript is complex. However, we do not have the same opinion of the reviewer regarding the fact that the methodology is not clearly documented, as each step is thoroughly explained in detail, and R packages, functions and function parameters are cited where appropriate. Having said that, we acknowledge that no general explanation of the pipeline was present in the methods. For this reason, we wrote a summary of the main pipeline as part of "Statistics and reproducibility" section in methods, free from technicalities, that will hopefully aid in the understanding of the mathematical framework and sequential protocol developed on the paper.

- Training and testing procedures of the model are not clear. They mentioned leave-one-out cross-validation, but it is difficult to know how validation and testing were performed.

R: We acknowledge that the evaluation of model performance was not explained clearly enough. The model was evaluated with an approximate leave-one-out method that does not require re-fitting the model with different training sets. We have stated this in the manuscript's methods (lines 535-537), and explicitly mentioned that the method is approximate where appropriate.

- Statistical testing is not well documented. How many samples? If t-test was used, did the data follow Gaussian distribution?

R: We agree with the reviewer that the explanation of how some statistical tests were performed was incomplete. We have corrected that where applicable (Figure 3 caption)

- The prediction performance was measured by ELPD. But there is no mathematical definition of it. Also, there should be error bars of ELPD and their statistical testing in Table 1

R: For data y_1, \dots, y_n , the expected log pointwise predictive density for a new dataset is defined as:

$$\sum_{i=1}^n \int p_t(\tilde{y}_i) \log p(\tilde{y}_i|y) d\tilde{y}_i$$

Being $p(\tilde{y}_i|y)$ the posterior predictive distribution, and $p_t(\tilde{y}_i)$ the distribution representing the true data-generating process for \tilde{y}_i . More details can be read in Vehtari et al., 2017 (10.1007/s11222-016-9696-4). Although we believe a formal mathematical definition within the text would not be appropriate, we followed the reviewer's advice and cited Vehtari et al. (2017) and R functions where applicable, so the readers have further information about the method.

We acknowledge that Table 1 was lacking a proper representation of the Standard Error. We added it as a new column in the table, and discuss its interpretation in the corresponding results section (lines 131-136).

- t-SNE generates different embedding when it is repeated. The author should assess the reproducibility of the model.

R: As the reviewer points out, t-SNE is stochastic, and thus, generates different landscapes when repeated. To test this as well as the differences across different t-SNE implementations, we applied the same pipeline of analysis (from 2D landscapes, to optimal landscape discretization, to predictions) to two completely different t-SNE, generated through different methods (barnes-hut and fast fourier interpolation t-SNE). Moreover, thanks to the reviewer's insights we have further assessed the reproducibility of the model using a UMAP embedding. Results comparing the three methods and their respective models can be found in figures S6 and S7.

Indeed, we run multiple versions of t-SNE embeddings within a single t-SNE implementation (i.e. Barnes-hut with perplexity = 639, 1% sample size). We found some variability in the shape of the landscape, as well as the distribution of sperm behaviours, as it occurs across different dimension reduction algorithms (i.e. Fit-SNE and UMAP). For purposes of simplicity and avoiding redundancy, we omitted this *within t-SNE* variability in the manuscript, as it is similar (although smaller) than the *across* dimension reduction methods. In any case, the variation observed basically served to convince us that our procedure and main conclusions are robust, always within the framework and methodological pipeline proposed.

REVIEWERS' COMMENTS:

Reviewer #1 (Remarks to the Author):

The authors have revised the paper in accordance with my concerns. I have no further comments.

Reviewer #3 (Remarks to the Author):

The authors satisfactorily addressed the reviewers' concerns. I also appreciate that the authors tested the variability of embeddings and demonstrated that their pipeline was robust. Therefore, I recommend it be published in Communications Biology.